# Measuring the Impact of the Multiple Cropping Index of Cultivated Land during Continuous and Rapid Rise of Urbanization in China: A Study from 2000 to 2015

**Ren Yang [1], Xiuli Luo [1], Qian Xu [2,\*], Xin Zhang [1] and Jiapei Wu [1]**

[1] School of Geography and Planning, Sun Yat-sen University, Guangzhou 510275, China; yangren666@mail.sysu.edu.cn (R.Y.); luoxli6@mail2.sysu.edu.cn (X.L.); zhangx697@mail2.sysu.edu.cn (X.Z.); wujp23@mail2.sysu.edu.cn (J.W.)

[2] School of Public Administration, Guangdong University of Finance & Economics, Guangzhou 510320, China

\* Correspondence: xuqian@gdufe.edu.cn

**Abstract:** With the continuous and rapid rise of urbanization in China, land use transition research has been carried out extensively. Multiple cropping is the content of land use recessive morphology research, and it is also a common agricultural system in China. Accordingly, further research on multiple cropping index (MCI) can enrich the land use transition research and help to evaluate China's food security. In order to examine the spatiotemporal changes and factors influencing the MCI of cultivated land in China, we collected MODIS remote sensing image data and land use classification data and conducted a remote sensing inversion on China's MCI from 2000, 2005, 2010, and 2015. The spatial distributions and evolution processes of the MCI were explored through spatial mapping, statistical analysis, and processing with the Geographic Information System; moreover, the influencing factors of MCI were explored quantitatively with principal component regression. The results were as follows: (1) at the provincial scale, the average MCI across Guangdong, Guangxi, Hainan, Henan, Anhui, and Jiangsu was high; meanwhile, the average MCI across Heilongjiang, Inner Mongolia, Ningxia, and Qinghai was low. Between 2000 and 2015, the number of provinces with low MCI increased gradually, and the average MCI decreased greatly in the southern provinces. (2) At the county scale, the Taihang Mountains, Qinling Mountains, and Hengduan Mountains formed the boundary of China's single cropping and multiple cropping indices. Dynamic changes in China's MCI were obvious, and the number of counties with MCI change values lower than 0 increased gradually. Last, (3) natural conditions, nonagricultural process, cultivated land quality, and agricultural intensification demonstrated different degrees of impact on the MCI; in particular, the influence of nonagricultural industries, pesticides, and agricultural plastic film on the MCI proved especially important. Future research should strengthen the existing work on related transformations in farmers' livelihoods, especially in terms of the return of rural labor force, the body of agricultural production, agricultural ecological issues, and the balance between increased crop production and reduced environmental pollution. In addition, agricultural policy design should pay more attention to cultivated land quality, the farmer who cultivates the land, and the multiple cropping potential of cultivated land.

**Keywords:** multiple cropping; cultivated land; land use transition; food security; influencing factors; China

## 1. Introduction

In 2015, the United Nations adopted the 17 Sustainable Development Goals, one of which is to eradicate hunger by 2030. However, the global demand for food production is continuing to rise due to population growth, diet changes, and increasing biofuel use [1]. Given the projected demand of our current course, global crop production must double by 2050 [2,3]; however, this is unlikely to happen, because climate change threatens farm

yields [4,5]. Indeed, several studies have shown that global grain production faces slow growth, stagnancy, or decline [3,6,7]. More specifically, four key global crops—maize, rice, wheat, and soybeans—demonstrate insufficient yield trends for doubling the global crop production by 2050 [4]. We need to ensure that the world's agricultural systems can produce enough food to feed the world's growing population; however, because our current agricultural systems are destroying the world's land, water, biodiversity, and climate, we must at once make our agriculture systems sustainable land use systems to reduce the adverse ecological impacts [8,9]. Future increases in food production should come from intensifying our use of existing land by halting agricultural expansion, improving cropping efficiency [10,11], closing "yield gaps" on underperforming lands, reducing waste, and shifting diets [8].

Asia's population is growing at a rate of 56 million people per year, putting increasing pressure on food demand and land use [12]. Recent studies suggested that urban expansion will result in a 1.8–2.4% loss of global croplands by 2030, with about 80% of the loss taking place in Asia and Africa [13]. China is the most populous country in Asia, with 19% of the world's population and only 8% of the world's croplands; notably, this asymmetry may affect China's—and, more broadly, the world's—food security [14,15]. Between the 1950s and 1990s, China's urbanization caused the rapid transition of cultivated land use, many cultivated lands transformed into nonagricultural uses, and rural areas served urban development [16,17]; accordingly, millions of rural agricultural laborers flowed into cities and nonagricultural industries. As a result, the proportion of China's rural population decreased from 82.08% in 1978 to 42.7% in 2016, and the proportion China's economy comprising the agricultural output decreased from 50.5% in the early 1950s to 8.6% in 2016 [18]. On the one hand, China's loss of its rural population led to the aging and hollowing of its rural areas [19,20]. On the other hand, it led to the serious abandonment of cultivated land, the weakened function of agricultural production, and the weakened status of farmers. For example, two million hectares of agricultural land fall out of production each year in China [14]. Despite rural depopulation, the land area of rural settlements has not decreased correspondingly. On the contrary, rural settlement areas nearly tripled from 1967 to 2008, and most of these increased areas occupied agricultural land, exacerbating the destruction of agricultural land [21]. In addition, China's agriculture has been characterized by "small-scale farming", "traditional farming methods", and "self-sufficiency" for a long time; therefore, low agricultural labor productivity has become a key weakness of agricultural competitiveness and sustainable development in China [22,23]. China already has the strictest cultivated land protection system, including land use planning; a basic cultivated land protection system; a land use regulation system; and a balanced system for the requisition, compensation, and consolidation of cultivated land and land consolidation. Although these policies have made remarkable achievements, they have not been able to address the shrinking of the country's cultivated land area and the deteriorating quality of regionally cultivated land. Today, the problem of the continuous nonagricultural, nongrain, and extensive use of cultivated land is worsening in China [16,24].

Some scholars claimed that increasing the farmland multiple cropping index (MCI) was the simplest method to enhance tillage [25–27]. Here, it is helpful to note that the cropping index refers to the number of times a crop is planted in a year, and a high MCI is related to an increase of the sown area, thus boosting food output [28]. Therefore, global crop production may be enhanced by agricultural intensification through multiple cropping to expand acreage without increasing the area of cultivated land [1].

Scholars from different countries have researched MCI in different regions of the world and focused on various aspects of the topic, including the evolution of Brazilian soybean double-cropping systems [29]; remote sensory monitoring of African cropping systems [30]; the relationship between crop rotation systems, water distribution, the political system, and the rational behavior of farmers in Egypt [31]; chemical weed control on crop yields in the Czech Republic [32]; and the effects of crop rotation on agrobiodiversity in Vanuatu [33]. Some scholars found that crop rotation and multiple cropping were used as methods

to change the physical properties of soil. For example, legumes can enhance the soil in a multiple crop lands by enhancing its organic carbon, fertility, aggregate stability, and vegetable yields under semi-arid conditions [34–36]. Meanwhile, scholarships have also been done to develop the first spatially explicit measure of the cropping intensity gap and, later, to uncover the differences between the potential and actual cropping intensity gaps; ultimately, these studies found that Latin America had a tremendous potential to expand its grain harvest by eliminating the cropping intensity gap, followed by Asia [1].

The multiple cropping system is a common farming practice in Asia, and agricultural production in Asia mainly emphasizes intensive multiple cropping production of rice and wheat [37]. Some researchers found that, in agricultural areas in Asia with large-scale irrigation systems, the land was mainly cultivated under the double-cropping system [38]. The current research on Asia focuses on India, Vietnam, and China. For example, scholarship done in India mostly used remote sensing satellite data to obtain crop rotation maps [39–41]; analyze the area and spatial distribution of multiple cropping crops [42]; and evaluate the efficiency and sustainability of India's cropping system by the MCI, area diversity index, and cultivated land utilization index [40]. In Vietnam, a time series of MODIS data was used to monitor the rice cropping intensity of the Mekong Delta [12]. For instance, based on MODIS time series imagery and field interviews, some studies explored the relationship between seasonal changes in the river and the temporal–spatial distribution of cropping systems and rice phenology in the Mekong Delta; they found that the change of the water environment was closely related to the changes of rice cropping systems [43,44].

Benefiting from a monsoon climate, China has one of the highest MCI in the world, with nearly 57% of its land cultivated using multiple cropping [45,46]. On the one hand, MCI is an important measure for evaluating China's food security, as its increase is essential to meet the growing demand for food [47,48]. On the other hand, MCI is an important evaluation index of land use transition research [49]. MCI researches the changes of recessive morphologies of land use transition and reflects the land management model by measuring the intensity level of the cultivated land use [50,51]. Therefore, it is of great practical significance to research MCI and its influencing factors to support national policy formulation in China [28,52,53]. In recent years, in order to evaluate the current situation of China's multiple cropping system, domestic scholars have actively researched the MCI, using different geographical scales. On the national scale, scholars have used various methods to calculate the actual MCI, the potential MCI, multiple cropping efficiency, and other indicators, such as the econometrics model [54], stochastic frontier analysis [28,47], Theil index [53], and continuous wavelet transform [55]; moreover, they have analyzed the spatiotemporal distribution of China's multiple cropping system and its regional differences. Xie and Liu discovered that China's MCI increased year after year from 1998 to 2012 [53]; meanwhile, Qiu et al. found that China's cropping intensity increased remarkably from 1982 to 1999 but declined slightly from 2001 to 2013 [55]. On the regional and provincial scales, the multiple cropping systems in some major agricultural production areas have been studied. For instance, Peng et al. found that Zhejiang Province's MCI decreased from 2001 to 2003 before increasing in 2004 and that the MCI in the southwest was higher than in the northeast [56]. Additionally, Zhang et al. extracted double-cropping systems in Northern China by using a Fourier analysis from the time series MODIS data [57]. Li et al. combined the results of field surveys and Landsat data and found that rice cropping systems in the Poyang Lake region showed an increasing trend from 2004 to 2010 [58], with an increased rate of about 20.2%. Feng et al. studied the effects of the planting patterns on the growth, yield, and economic benefits of cotton in a wheat–cotton double-cropping system versus monoculture cotton [59].

Previous research on the MCI mainly focused on a single scale in China, and it seldom considered the whole country through a multi-scale perspective. It focused on how to improve the technology and technical aspects to extract MCI with remote sensing data, as well as the spatiotemporal distribution characteristics; however, there is insufficient

research on the regional differences and driving forces. Cultivated land is an important land type in the research of rural land use transition, and it is closely related to human production and life. Cultivated land use reflects the evolution of human–environment relationship in rural areas and also reflects the current situation and problems of China's agriculture and rural society. Therefore, it is necessary to have a comprehensive understanding of the national MCI. Based on the limitations of the previous research, this study sought to answer the following questions: What are the characteristics of the dynamic evolution and spatiotemporal distribution of MCI? Where are these characteristics distributed at the provincial and county scales? What accounts for the differences in the spatiotemporal distributions?

## 2. Research Design

### 2.1. Research Structure

This research assumed "phenomenon–problem–pattern–process–mechanism" as the main logical structure, and its contents mainly included the research foundation, scientific problems, data processing, spatiotemporal analysis, and influencing factor analysis; accordingly, the detailed research process for each part was designed as follows (Figure 1).

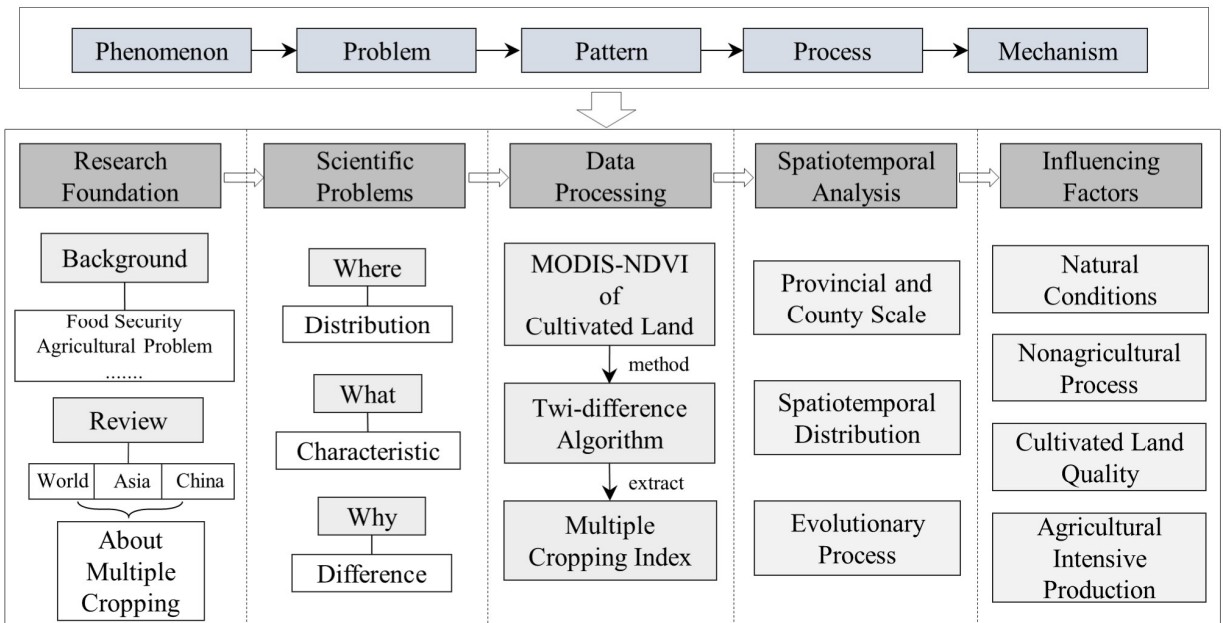

**Figure 1.** Framework for examining the spatiotemporal characteristics and influencing factors of the MCI.

This paper takes the human–land relationship as its core principle and built a research framework concerning the influencing factors starting from four aspects: natural conditions, nonagricultural process, cultivated land quality, and agricultural intensive production (Figure 2).

First, the natural conditions of the agricultural climate conditions, topography, landform, and soil have deeply determined the cropping system of traditional agriculture. Second, the urban–rural transition has dramatically changed the interactions between urban and rural production; notably, following the acceleration of rural economic development, significant changes took place in rural areas—for example, rural labor forces, industrial structures, and employment structures began to move away from agriculture. Third, cultivated land quality affects the increase of MCI and is at once affected by human activities and natural disasters. Human activities include the use of agricultural fertilizers, pesticides, and agricultural films. Finally, agricultural modernization and the agricultural production efficiency directly impact intensive agricultural production, thereby changing the MCI. Agricultural modernization is influenced by factors such as traffic, irrigation,

the conditions of agricultural machinery, and investment level. As the main body of agricultural production, farmers' input and enthusiasm for farming determine the level of agricultural production efficiency, which is also reflected in per capita cultivated land area and grain per labor. Moreover, because farmers are rational, economic individuals, their enthusiasm is directly affected by their agricultural income, which also influences the MCI. Based on the above analysis and the availability of data, 19 indexes were selected to investigate the factors impacting the MCI (Table 1).

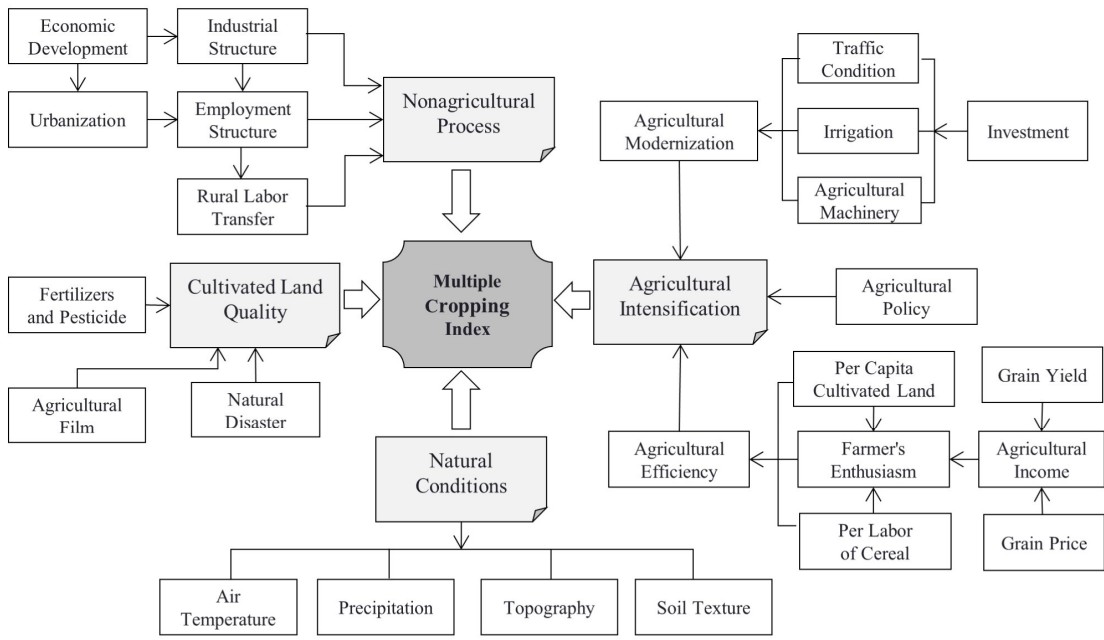

**Figure 2.** The research framework concerning the influencing factors.

**Table 1.** The influencing factors index of the MCI in China.

| Types | Indexes | | Calculation Method and Data Description |
|---|---|---|---|
| Nonagricultural process | Nonagricultural population | $x1$ | Nonagricultural population/Total population |
| | Nonagricultural industry | $x2$ | GDP of secondary and tertiary industry/GDP |
| | Urbanization rate | $x3$ | China Statistical Yearbook |
| | Per capita GDP | $x4$ | China Statistical Yearbook |
| Cultivated land quality | Density of agricultural fertilizer | $x5$ | Agricultural fertilizer/Agricultural acreage |
| | Density of pesticide | $x6$ | Pesticide/Agricultural acreage |
| | Density of agricultural plastic film | $x7$ | Agricultural plastic film/Agricultural acreage |
| | Natural disaster | $x8$ | Natural disaster/Crop sown area |
| Agricultural efficiency | Cultivated area per capita | $x9$ | Cultivated land area/Rural population |
| | Grain yield per unit area | $x10$ | Grain total yield/Cultivated land area |
| | Grain yield per labor force | $x11$ | Grain total yield/Agricultural population |
| | Farmers' net income per capita | $x12$ | China Rural Statistical Yearbook |
| Agricultural modernization | Irrigation rate | $x13$ | Irrigated area of cultivated land/Cultivated land area |
| | Agricultural machinery per unit area | $x14$ | Agricultural machinery/Agricultural land area |
| | Investment conditions | $x15$ | Total investment in fixed assets/Administrative area of land |
| Natural conditions | Average annual precipitation | $x16$ | Resource and Environment Data Cloud Platform |
| | Average annual temperature | $x17$ | Resource and Environment Data Cloud Platform |
| | Soil Texture | $x18$ | Resource and Environment Data Cloud Platform |
| | Relief degree of land surface | $x19$ | Global Change Data & Discovery |

*2.2. Methods and Procedures*

2.2.1. Extraction Methods of the MCI

The MCI refers to the times of sequential crop planting in the same cultivated land in one year, reflecting the utility degree of arable land to be used at a certain time. It is desirable to extract the MCI and its spatial distribution information by remote sensing

image data. In this study, the arable land coverage regions of the MODIS-NDVI remote sensing image data were extracted from the land use classification data for 2000, 2005, 2010, and 2015. The extraction methods of the MCI were as follows [56,60].

As shown in Figure 3, a coordinate system was established, with time as an x-coordinate and normalized difference vegetation index (NDVI) as a y-coordinate, showing a periodic dynamic of crops in the sowing, germinating, earring, and harvesting stages. Specifically, Figure 3a shows crops' performances under the single-cropping system (one-peak curve). Figure 3b shows crops' performances under the double-cropping system (two-peak curve). Figure 3c illustrates crops' performances under the triple-cropping system (triple-peak curve). Last, Figure 3d concerns bare land or abandoned land, characterized by insignificant peaks and low NDVI. Hence, the number of NDVI peaks within a year can be understood as the MCI of the area.

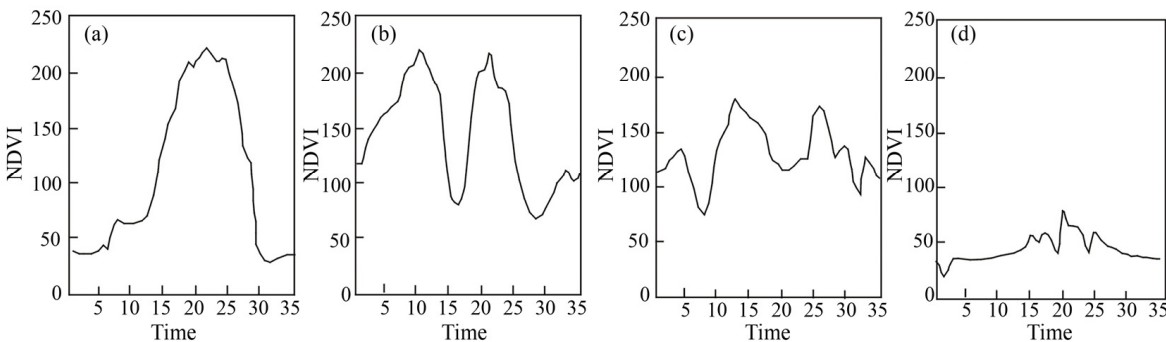

**Figure 3.** The NDVI time series curve smoothing by HANTS. Note: time as an x-coordinate and NDVI as a y-coordinate, without dimensionality. (**a**) Crops' performances under the single-cropping system (one-peak curve). (**b**) Crops' performances under the double-cropping system (two-peak curve). (**c**) Crops' performances under the triple-cropping system (triple-peak curve). (**d**) Crops' performances under bare land or abandoned land, characterized by insignificant peaks and low NDVI.

Considering the relationship between the NDVI time series curve and the MCI, it can be concluded that the calculation of the MCI is equal to the extraction process of the peak frequency of the NDVI time series. The extraction formula is as follows:

$$F_i = \frac{S_{sumapex_i}}{S_{sumpixel_i}} \tag{1}$$

where $F_i$ represents the frequency of NDVI time series peaks of the administrative units providing the MCI. $S_{sumapex_i}$ is the total number of peaks formed within a year on all pixel values. $S_{sumpiexl_i}$ refers to the total number of pixels in the NDVI curves. As for the frequency of the peaks, it was extracted using the Twi-difference algorithm, for the NDVI time series of each pixel can be considered as a sequence of the discrete points. Specifically, the differences of the neighboring NDVI values should be computed with Formula (2) to conclude $S_1$ (Sequence 1), and the plus or minus would be determined by Formula (3). If the value is positive, 1 is recorded; alternatively, $-1$ is recorded to yield $S_2$ (Sequence 2). Finally, the differences of $S_{2i}$ and $S_{2i+1}$ can be calculated using Equation $S_3$ (Sequence 3).

$$S_{1_i} = NDVI_{i-1} - NDVI_i \tag{2}$$

$$S_2 = \begin{cases} 1 \ S_{1_i} > 0 \\ -1 \ S_{1_i} < 0 \end{cases} \tag{3}$$

$$S_{3_i} = S_{2_i} - S_{2_{i+1}} \tag{4}$$

In all the formulas above, *i* represents the *i*th pixels in different sequences. It is concluded that the peak of crop growth occurred in $S_3$, where the pixel value was $-2$, and

the pre and post values were both 0. MODIS-NDVI data from 16 consecutive days in 2000, 2005, 2010, and 2015, temporally and spatially based on the spatial vector data of the land use type, was analyzed using MATLAB and ArcGIS; the NDVI peaks' frequency of each raster cell is shown in Figure 4—the inversion of the MCI was accomplished. Next, the provincial and county administrative units were used to carry on the statistics related to the MCI and the MCI means of Chinese counties and provinces were presented using the ArcGIS10.2 platform to prepare them for the subsequent analysis.

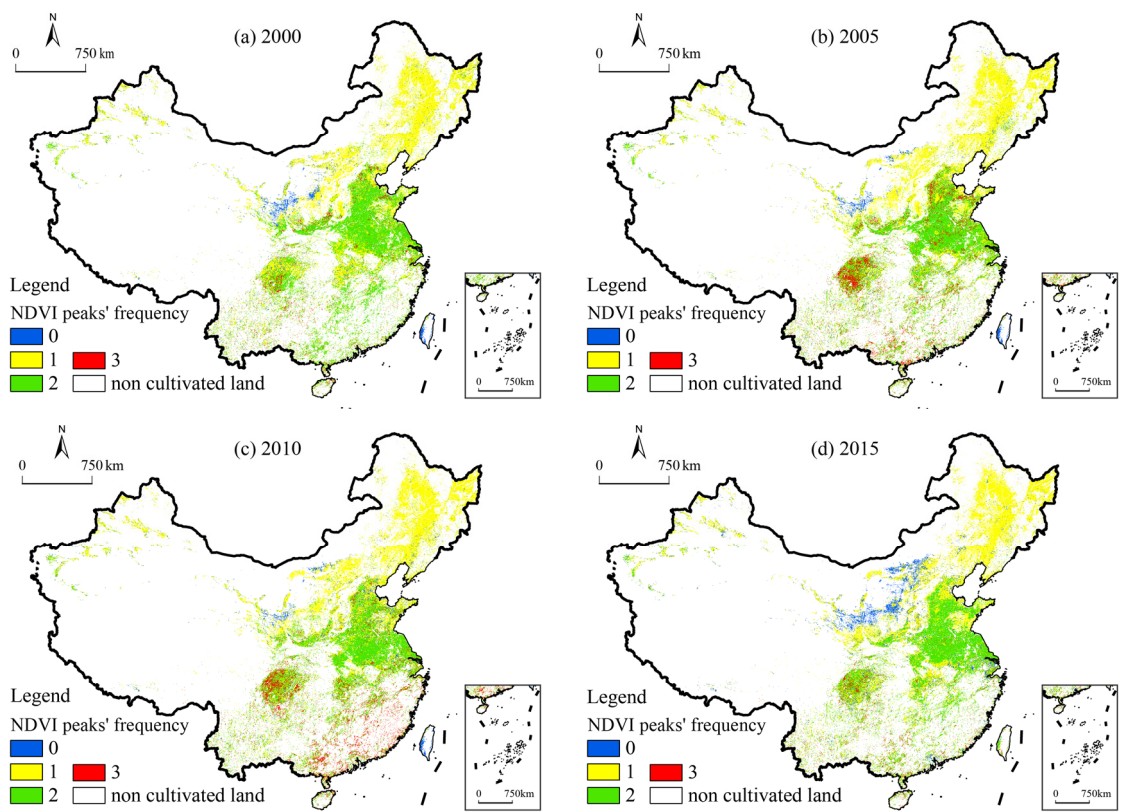

**Figure 4.** Spatial distribution of the NDVI peak frequency of each raster cell in different years in China. (**a**) NDVI peak frequency in 2000. (**b**) NDVI peak frequency in 2005. (**c**) NDVI peak frequency in 2010. (**d**) NDVI peak frequency in 2015.

### 2.2.2. Principal Component Regression

Pearson first proposed the principal component analysis method [61], which was popularized and developed by Hotelling [62]. Later, Massy proposed principal component regression based on the idea of the principal component analysis [63]. The core idea of principal component regression is to transform multiple independent variables into a few principal components through dimension reduction to eliminate collinearity among original independent variables; without changing the original independent variables' interpretation of the dependent variables, the transformed principal component is used to replace the original independent variable for the regression analysis.

The years 2000 and 2015 were selected to explore the factors influencing the MCI from 2000 to 2015, and the research sample was taken from 31 provinces of mainland China. Due to the large number of indicators but small sample sizes, there was severe multicollinearity among the independent variables; thus, the principal component regression was used to increase the reliability of the results. The MCI was the dependent variable, and the $x1$–$x19$ indicators in Table 1 were the independent variables. To make the indicators comparable, the deflator index of each province was used to eliminate the interference of the price factors for the economic indicators, such as per capita GDP, total investment in fixed assets, and farmers' net income per capita. The processed data were analyzed by principal

component regression using SPSS18.0 (https://www.ibm.com/analytics/spss-statistics-software) (accessed on 21 October 2019).

First, the factor analysis tool of SPSS18.0 was used for the principal component analysis, and Kaiser–Meyer–Olkin (KMO) and Bartlett's tests of sphericity were used as the criteria for the applicability judgment. The results showed that the KMO in 2000 and 2015 were 0.656 and 0.646, and both met the test criteria. Bartlett's test of sphericity met the significance level. Hence, the samples selected in this research were suitable for the principal component analysis, because both conditions met the applicability requirements. Next, the principal components were determined by the basic principle that the cumulative variance rate was more than 85% and the eigenvalue was close to 1. As shown in Table 2, five principal components were confirmed in 2000 and 2015, respectively. The cumulative variance rate for 2015 was 84.10%, close to 85%, and the eigenvalue of the sixth principal component differed significantly from 1; therefore, five principal components were retained in 2015. There was no collinearity between the five new independent variables.

**Table 2.** Principal component eigenvalues and total variances for 2000 and 2015.

| Component | 2000 | | 2015 | |
|---|---|---|---|---|
| | Eigenvalue | % of Cumulative Variance | Eigenvalue | % of Cumulative Variance |
| 1 | 7.84 | 41.25 | 7.57 | 39.82 |
| 2 | 5.04 | 67.78 | 4.00 | 60.86 |
| 3 | 1.61 | 76.26 | 1.96 | 71.19 |
| 4 | 1.24 | 82.81 | 1.54 | 79.32 |
| 5 | 0.93 | 87.69 | 0.91 | 84.10 |

Second, by multiplying the factor score and the square root of the eigenvalue resulting from the principal component analysis, the principal component score was obtained as the new independent variable score. Next, the dependent variable—namely, the MCI—was standardized by the z-score method, and the standardized MCI and five principal component scores were analyzed by linear regression. The results are shown in Table 3.

**Table 3.** Principal component regression results in 2000 and 2015.

| Type | 2000 | | 2015 | |
|---|---|---|---|---|
| | Unstandardized Coefficients (*B*) | Sig. | Unstandardized Coefficients (*B*) | Sig. |
| (Constant) | 0.00 | 1.00 | 0.00 | 1.00 |
| Principal Component 1 | 0.24 ** | 0.00 | 0.19 ** | 0.00 |
| Principal Component 2 | −0.24 ** | 0.00 | −0.34 ** | 0.00 |
| Principal Component 3 | −0.01 | 0.92 | 0.15 ** | 0.03 |
| Principal Component 4 | −0.14 | 0.12 | 0.07 | 0.37 |
| Principal Component 5 | 0.12 | 0.26 | 0.02 | 0.85 |

(Note: ** represents sig. < 0.05).

The results show that Principal Component 1 ($F_1$) and Principal Component 2 ($F_2$) reached significant levels (sig. < 0.05) in 2000, and Principal Component 1 ($F_1'$), Principal Component 2 ($F_2'$), and Principal Component 3 ($F_3'$) reached significant levels (sig. < 0.05) in 2015. Thus, we put the unstandardized coefficients (*B*) into the formulas $F_{2000} = B_1 F_1 + B_2 F_2$ and $F_{2015} = B_1' F_1' + B_2' F_2' + B_3' F_3'$. Next, we divided the component matrix in tSPSS18.0 (principal component analysis) by the square root of the eigenvalue to get the eigenvector ($a_i$) of each principal component before replacing $a_i$ in the principal component formula $F_k = a_{1k} Z_1 + a_{2k} Z_2 + \ldots + a_{nk} Z_n$ (Z is the standard matrix of the independent variable). Finally, by substituting $F_k$ into formulas $F_{2000}$ and $F_{2015}$, the normalized



independent variable $X$ and dependent variable $Y$ were reduced to the original data $y$ and $x$, and the final regression formulas $y_{2000}$ and $y_{2015}$ were obtained.

### 2.3. Data Sources

First, MODIS-NDVI remote sensing image data and vector data in 2000, 2005, 2010, and 2015 on the spatiotemporal distributions of the agricultural lands were provided by the Earth System Scientific Data Sharing Platform, Chinese Academy of Science (data source of Figure 4). Second, the indexes of the influencing factors included information on the economic, social, and natural conditions. The economic and social data, such as population, gross domestic product, cultivated land area, and grain output, were derived from the China Statistical Yearbooks (2001 and 2016), China Rural Statistical Yearbooks (2001 and 2016), China Population & Employment Statistics Yearbooks (2001 and 2015), and other statistical yearbooks for the provinces (data sources of Table 1 and Figure 5). In particular, the data for China's agricultural population and nonagricultural population were only available until the end of 2014 (China Population & Employment Statistics Yearbook 2015); therefore, the data for 2015 were replaced by those from the end of 2014. Data on China's cultivated land area were obtained from China's Land and Resources Bulletin (data source of Table 1 and Figure 5). Third, the natural condition data, such as soil texture, annual average temperature, and annual average precipitation, were obtained from the Resource and Environment Data Cloud Platform, Chinese Academy of Science (data source of Table 1). Data sources for the soil texture include the percentages of sand, silt, and clay. According to the China soil texture classification (1985) [64], the soil texture data were divided into three types: clay, loam, and sand. They were marked as 1, 2, and 3, respectively, in the regression analysis. Besides, the relief degree of the land surface was derived from the research data of You et al. [65], who shared it in the Global Change Research Data Publishing & Repository (data source of Table 1). Finally, the deflation index used to process the economic data, the calculation process involved in the GDP, and the GDP index were taken from the China Statistical Yearbook (data source of Table 1).

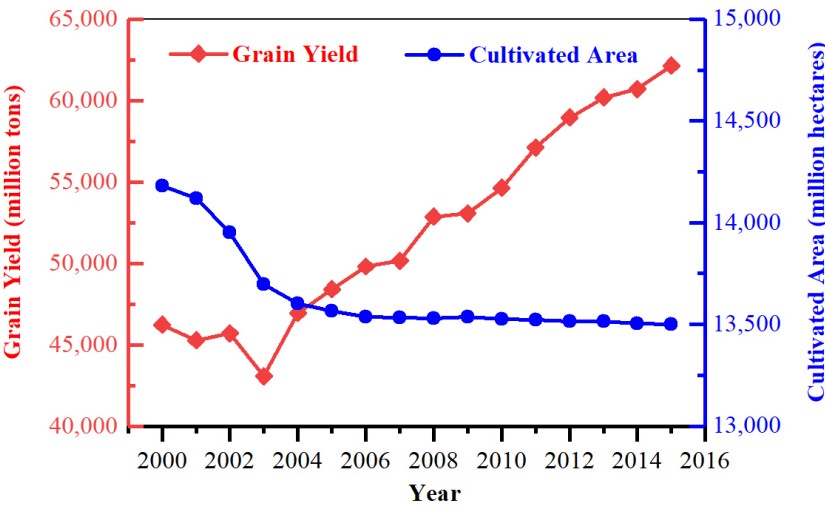

**Figure 5.** China's cultivated land areas and grain yields, 2000–2015.

In 2000, 2005, 2010, and 2015, mainland China did not suffer from severe agricultural disasters; therefore, it is reasonable to choose these four years as representatives to study the MCI from 2000 to 2015 without much contingency. In addition, due to the limited availability of the data, this research only studied mainland China and did not involve Hong Kong, Macao, or Taiwan.

## 3. Results

### 3.1. Cultivated Land Area and Grain Yield

The cultivated land area in China from 2000 to 2008 was reported by the first national land survey and the cultivated land area from 2009 to 2015 by the second national land survey. However, the criteria and techniques used in the two surveys were different, and the values between them could not be simply compared. Therefore, referring to the research of Chen et al. [66], this paper adjusted the cultivated land area from 2000 to 2008 to the cultivated land area based on the second survey, as shown in Figure 5.

In general, China's cultivated land areas showed a downward trend from 2000 to 2015, from 141.83 million hectares in 2000 to 135.00 million hectares in 2015, with an average annual decline of 0.46 million hectares. Although China's cultivated land area had been declining over the past 15 years, it did not break the "red line of 1.80 billion mu of cultivated land in China". The changes in grain yields fluctuated. Overall, the grain yield showed an upward trend, rising from 462.18 million tons in 2000 to 621.44 million tons in 2015, with an average annual growth of 10.62 million tons. Notably, the grain yields from 2000 to 2001 and from 2002 to 2003 showed a declining trend, and the grain yield in 2003 was the lowest in the past 15 years; meanwhile, the grain yields from 2003 to 2015 showed an upward trend with a relatively fast rate of increase and an average annual increase of 15.90 million tons.

### 3.2. Spatiotemporal Change of MCI at the Provincial Scale

Figure 6 shows the cultivated land MCI of 31 provinces in China in 2000, 2005, 2010, and 2015. The research results showed that the MCI varied dramatically in different provinces. Notably, the MCI was high in the coastal provinces of Southern China. For example, Guangdong and Guangxi had the highest average MCI (190%) from 2000 to 2015, followed by Hainan (187%). In 2005 and 2010, the MCI in Guangdong exceeded 200%, and Fujian, Guangxi, and Jiangxi had more than 200% MCI in 2010; meanwhile, the MCI of Hainan, Guangxi, Henan, Anhui, and Jiangsu were high and stable above 160% from 2000 to 2015. Most of these areas with high MCI are located in the south of China, which is characterized by a subtropical monsoon climate and good water and heat conditions conducive to the multi-cropping of the cultivated land. Moreover, Henan, Jiangsu, and Guangdong are among China's major agricultural provinces and, thus, areas where the development of agriculture is particularly supported. Along these lines, their level of agricultural modernization is high, which makes their MCI higher and the rate of change relatively stable. Most the provinces with smaller MCI are located in the northeast and northwest. From 2000 to 2015, the MCI of the provinces of Heilongjiang, Inner Mongolia, Ningxia, and Qinghai were lower than 100%, and the number of provinces with MCI lower than 100% increased year after year, from five provinces in 2000 to 11 provinces in 2015. The Shaanxi, Shanxi, and Gansu Provinces were newly added to this list in 2015.

This paper divided the 2000–2015 interval into three stages to analyze the changes in the MCI of the cultivated land (Figure 7). It was concluded that the number of provinces with a declining MCI increased with time and that the declining range of the MCI increased. An MCI of 61.29% of the provinces showed an increasing trend from 2000 to 2005; in Sichuan, Guangdong, and Guangxi, the MCI increased by 50.38%, 25.52%, and 23.59%, respectively. The most rapid decline was in the economically developed city of Beijing, which decreased by 30.59%. Generally, an MCI of 51.61% of the provinces showed a trend of growth from 2005 to 2010. Meanwhile, compared with the previous stage, the number of provinces with a declining MCI increased. Shandong and Hebei had the fastest decline in their MCI at 25.03% and 23.12%, respectively. The southern provinces, mainly Chongqing, Fujian, Jiangxi, Zhejiang, Guangdong, and Hunan, had rapid growth rates of the MCI, with Fujian showing the fastest growth rate at 75.58%. The change in the MCI from 2010 to 2015, as shown by the green line in Figure 7, was almost lower than 0. In fact, the MCI in 93.55% of the provinces showed a downward trend, while the MCI of Shandong and Hebei showed a small increase, with 8.34% and 0.54%, respectively. In the southern provinces,

the MCI decreased by a large margin, and the five provinces of Fujian, Guangdong, Jiangxi, Zhejiang, and Shanghai showed decreases of more than 50%. From the above results, the dynamic changes of the MCI of the southern provinces were obvious. Before 2010, China's MCI was in a growth trend, but this was replaced by a downtrend and substantial decline after 2010. This trend is not conducive to China's food security, because most of the double-cropping and multi-cropping rice areas in China are located in its southern regions.

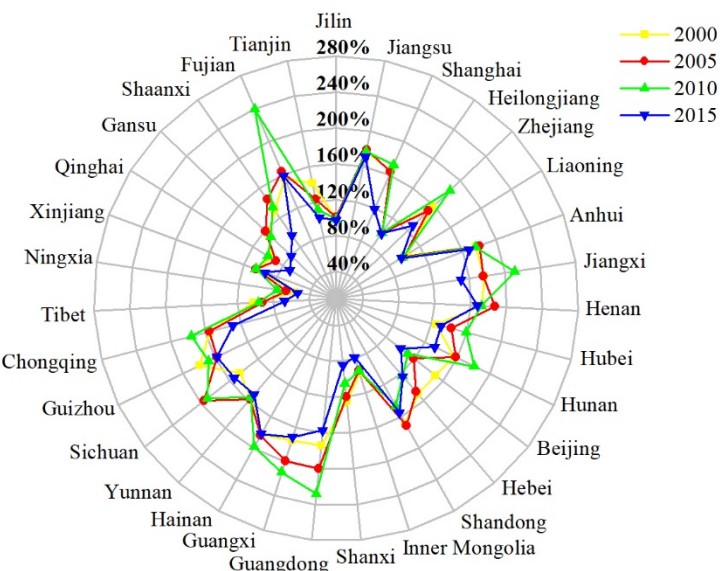

**Figure 6.** The value of the MCI in Chinese provinces, 2000–2015.

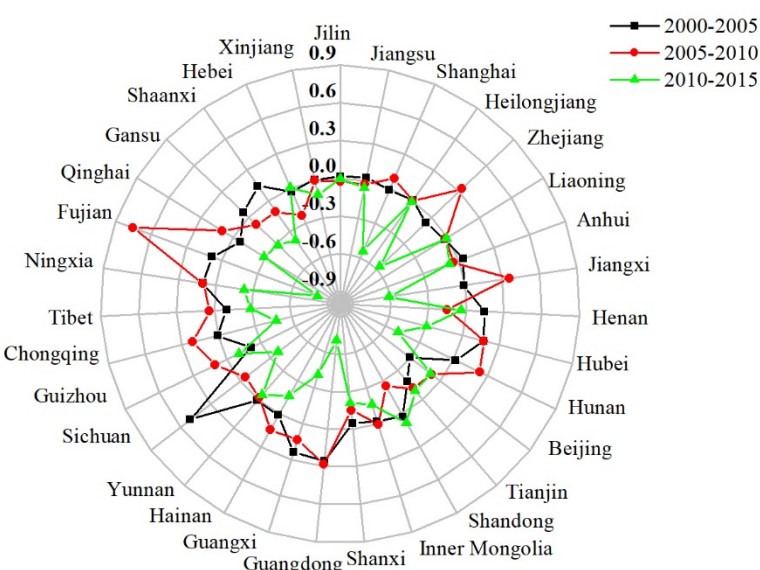

**Figure 7.** The changes in the values of the MCI in Chinese provinces from 2000 to 2005, 2005 to 2010, and 2010 to 2015.

### 3.3. Spatiotemporal Change of MCI at the County Scale

Figure 8 shows the spatial distribution of China's MCI from 2000 to 2015, and Figure 9 illustrates the statistics of the MCI of counties across different delimited ranges. China's MCI was roughly divided into a single-cropping system and a multiple-cropping system by the "Taihang Mountains–Qinling Mountains–Hengduan Mountains" boundary from 2000 to 2015 (Figure 8). The MCI in the north of the boundary was less than 100% and more than 100% in the south.

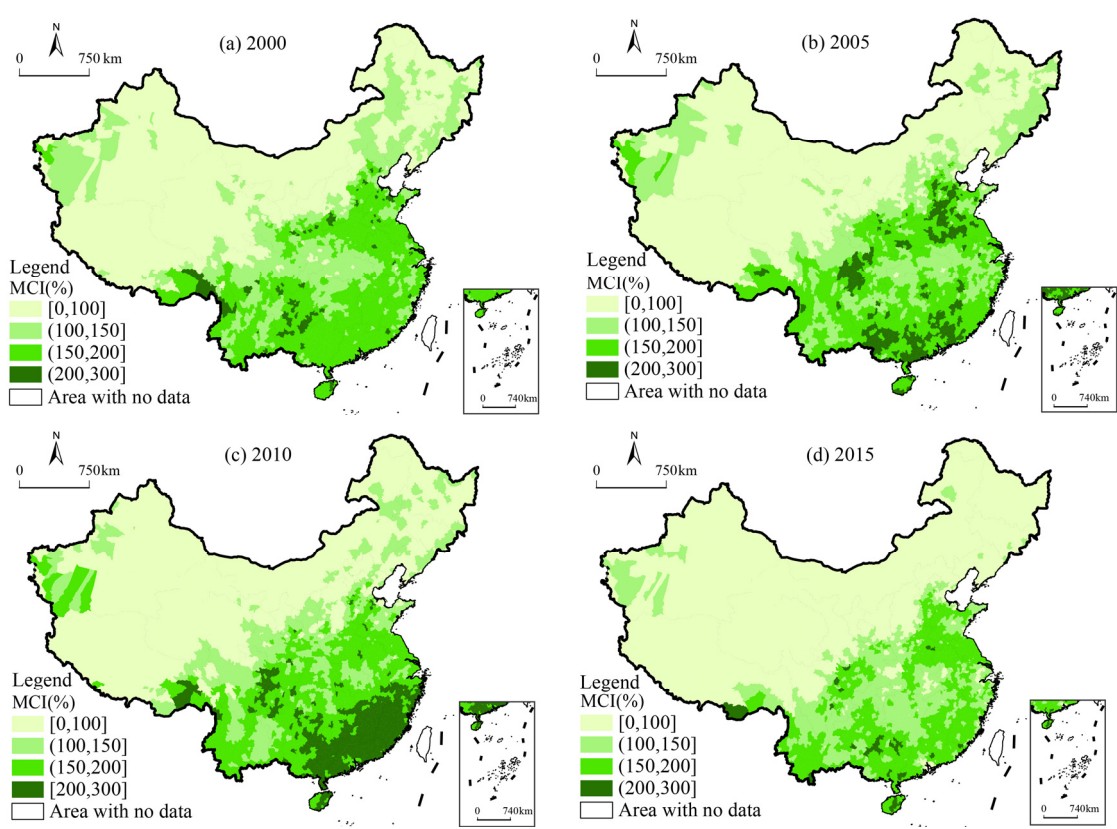

**Figure 8.** The spatial distributions of the MCI in counties in different years. (**a**) The MCI in counties in 2000. (**b**) The MCI in counties in 2005. (**c**) The MCI in counties in 2010. (**d**) The MCI in counties in 2015.

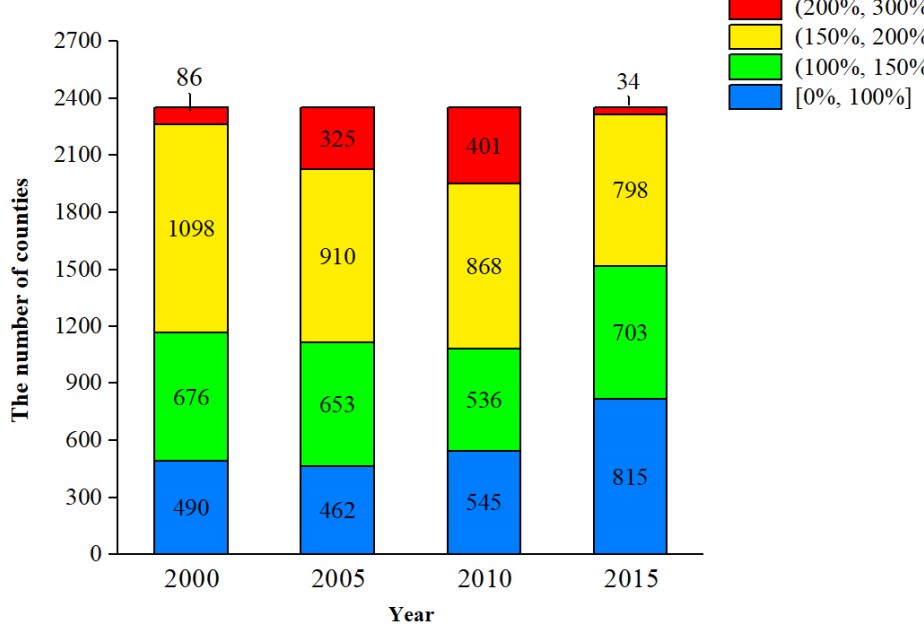

**Figure 9.** The number of counties demonstrating each MCI range.

In 2000, the average value of China's MCI was 143.16%. The number of counties with MCI in the 150–200% range was the largest, with a total of 1098 units, accounting for 46.72% of China's counties (Figure 9); notably, these counties were primarily located in the East China region, South China region, and Yunnan Province (Figure 8a). Meanwhile, the number of counties with MCI in the 100–150% range was 676, accounting for 28.77%

of China's counties; these counties were mainly distributed in the provinces of the mid-Yangtze River, Northeast China, and Southwest Xinjiang. The number of counties with MCI in the 0–100% range was 490, accounting for 20.85% of China's counties; these counties were mainly distributed in the north of the boundary of "Taihang Mountains–Qinling Mountains–Hengduan Mountains". Only 3.66% of the counties demonstrated an MCI greater than 200%, and the spatial distribution was scattered.

In 2005, the average value of China's MCI was 148.66%, an increase from 2000. In particular, the number of counties with MCI in the 200–300% range increased to 325, accounting for 13.83% of China's counties (Figure 9); these counties were mainly distributed in the North China region, South China region, and Chengdu Plain (Figure 8b). Counties with MCI in the 150–200% range still had the highest proportion but decreased to 38.72% of China's counties; these counties were primarily distributed in the North China Plain and to the south of the Yangtze River. The spatial distribution with MCI lower than 150% was similar to that observed in 2000.

In 2010, the average value of China's MCI was 152.03%, an increase from 2005. The number of counties with MCI in the 200–300% range continuously increased to 401, accounting for 17.06% of China's counties (Figure 9); these counties were mainly distributed in the hilly region of Southeast China and Chengdu Plain (Figure 8c). The number of counties with MCI in the 100–200% range accounted for 66.51%, which were mainly distributed in the south of the boundary of "Taihang Mountains–Qinling Mountains–Hengduan Mountains" and some areas in the southwest of Xinjiang Province and Northeast China. The areas where the MCI was lower than 100% were mainly distributed in the north of the boundary of "Taihang Mountains–Qinling Mountains–Hengduan Mountains".

In 2015, the average value of China's MCI was 125.95%. Compared to 2010, the average value of the MCI decreased by 26.08% and 5.22% annually. The spatial distribution of MCI in the 200%–300% range decreased sharply to 34 counties (Figure 9) that were scattered across the southern region (Figure 8d), accounting for 1.45%. The number of counties with MCI in the 0–100% range increased sharply, mainly distributed in the north of the boundary of "Taihang Mountains–Qinling Mountains–Hengduan Mountains". However, in some counties south of the boundary, the MCI was also lower than 100%, especially in the economically developed Pearl River Delta and Yangtze-Huaihe regions, and the MCI of many counties in Hubei Province were also lower than 100%. The number of counties with MCI in the 100–150% range increased significantly to 703, accounting for 29.91% of China's counties; these counties were mainly distributed in the provinces of the middle and lower reaches of the Yangtze River and Yunnan. The number of counties with MCI in the 150–200% range decreased to 798, accounting for 33.96% of China's counties; these counties were mainly distributed in Eastern China, Southern China, and southwest of Yunnan Province.

Algebraic operations were carried out for the four years between 2000 and 2015, and the variations in the MCI values for the four periods—2000–2005, 2005–2010, 2010–2015, and 2000–2015—were obtained (Figure 10).

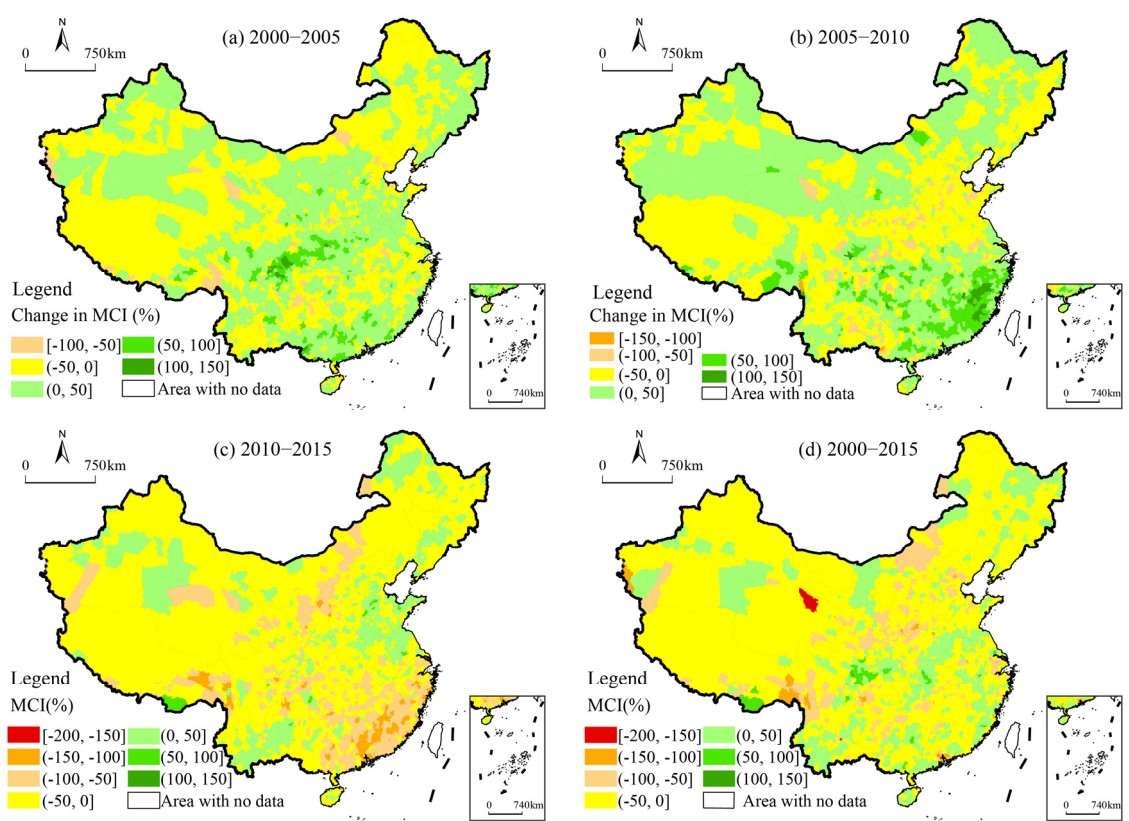

**Figure 10.** Spatial distributions of the MCI value changes across Chinese counties. (**a**) The MCI value changes from 2000 to 2005. (**b**) The MCI value changes from 2005 to 2010. (**c**) The MCI value changes from 2010 to 2015. (**d**) The MCI value changes from 2000 to 2015.

From 2000 to 2005, 56.17% of China's counties demonstrate variations in the MCI higher than 0; these counties were mainly distributed across the central provinces, Xinjiang, three northeastern provinces, and the coastal provinces of Guangdong and Guangxi (Figure 10a). The rates of change in the MCI values of most counties were under 50%, while changes greater than 50% were evident in only 6.17% of the counties—these counties were distributed across the areas of the Chengdu Plain, Guangxi, and Guangdong. Meanwhile, 43.83% of the counties showed a change in the MCI values lower than 0; these counties were widely distributed across the northeast region, northwest region, Tibet, Yunnan, and Guizhou. Only 2.26% of the counties demonstrated a variation in value in the range between −200% and −50%, and their distribution was relatively scattered.

From 2005 to 2010, 49.23% of the counties exhibited a variation in the MCI value greater than 0, with a decrease of 6.94% compared with the previous stage; the counties were mainly distributed across the southeastern provinces, northwest, and northeast regions (Figure 10b). The number of counties in which the MCI variation values were greater than 50% increased, accounting for 9.79% of the counties; these counties were mainly distributed in Guangdong, Fujian, Jiangxi, and Zhejiang. In contrast, 50.77% of the counties demonstrated a change in MCI values lower than 0; this was higher than the percentage of counties with a change higher than 0; notably, these counties were mainly distributed across the north of the Qinling–Huaihe River Line, the south of the Inner Mongolia Autonomous Region, and—widely—in Tibet. Moreover, the number of counties with a change in MCI values in the range between −200% and −50% increased, accounting for 4.81%, and the distribution remained scattered.

From 2010 to 2015, the proportion of counties with a change in MCI values higher than 0 dropped sharply to 20.47%, and the proportion of counties with changes in the MCI lower than 0 was almost four times that of the proportion of counties with changes in the MCI higher than 0 (Figure 10c). In terms of spatial distribution, the MCI changes across

the cultivated land in China were generally lower than 0. More precisely, the proportion of counties with variations in the MCI in the range between −200% and −50% increased sharply to 23.53%, and these counties were mainly distributed across China's southeast coastal provinces. However, only the Bohai Rim region, Heilongjiang, Yunnan, and Xinjiang showed MCI changes higher than 0, of which only 0.68% of the county MCI were higher than 50%.

Overall, from 2000 to 2015, MCI changes greater than 0 were evident in 24.55% of the Chinese counties, and the number of counties with MCI changes lower than 0 was more than three times that of the number of counties with values higher than 0 (Figure 10d). During these 15 years, China's MCI showed a significant downward trend—in particular, the MCI declined greatly in the three provincial border regions of Yunnan, Sichuan, and Tibet; some provinces in the upper and middle reaches of the Yellow River basin; the Beijing–Tianjin–Hebei region; the Yangtze River Delta; and the Pearl River Delta, with variation values in the range between −200% and −50%. Conversely, the districts of counties with MCI higher than 0 were scattered, mainly across Northeast China, the Chengdu Plain, Jianghan Plain, Henan Province, Xinjiang Province, the Western Inner Mongolia Autonomous Region, and the southeast coastal provinces. The change value was only 1.53% above 50%, which was distributed only in Southern Tibet, the Chengdu Plain, Jianghan Plain, and other regions.

### 3.4. Influencing Factors of MCI

The final regression formula is $y = \sum\limits_{i=1}^{n} b_i x_i + c$ ($n = 19$), $c$ is the constant, and $b_i$ is the coefficient of the influencing factor ($x_i$). The results of $b_i$ and $c$ are shown in Table 4. Specifically, the size of $b_i$ reflects the degree of influence of this factor on the MCI, and the positive and negative values of $b_i$ reflect the effect of this factor on the MCI.

**Table 4.** Principal component regression results of the influencing factors for the MCI.

| Type | | 2000 | | 2015 | |
|---|---|---|---|---|---|
| Index | $x$ | Coefficient: $b$ | Constant: $c$ | Coefficient: $b$ | Constant: $c$ |
| Non-agricultural population | $x1$ | −0.0025 | | −0.0019 | |
| Non-agricultural industry | $x2$ | −0.0543 | | −0.6093 | |
| Urbanization rate | $x3$ | −0.0007 | | −0.0011 | |
| Per capita GDP | $x4$ | $5.5 \times 10^{-7}$ | | $-8.2 \times 10^{-7}$ | |
| Density of agricultural fertilizer | $x5$ | 0.3645 | | 0.2152 | |
| Density of pesticide | $x6$ | 9.7628 | | 3.7945 | |
| Density of agricultural plastic film | $x7$ | 1.4343 | | −0.4621 | |
| Natural disaster | $x8$ | −0.5069 | | −0.4206 | |
| Cultivated area per capita | $x9$ | −1.0038 | | −0.0876 | |
| Grain yield per unit area | $x10$ | 0.0354 | 0.1599 | 0.0233 | 1.7466 |
| Grain yield per labor force | $x11$ | −0.1731 | | −0.0140 | |
| Farmers' net income per capita | $x12$ | $1.7 \times 10^{-5}$ | | $-4.9 \times 10^{-7}$ | |
| Irrigation rate | $x13$ | 0.1615 | | 0.0351 | |
| Agricultural machinery per unit area | $x14$ | 0.0047 | | 0.0044 | |
| Investment conditions | $x15$ | $3.0 \times 10^{-5}$ | | $-7.0 \times 10^{-6}$ | |
| Average annual precipitation | $x16$ | $9.2 \times 10^{-6}$ | | $1.8 \times 10^{-5}$ | |
| Average annual temperature | $x17$ | 0.0014 | | 0.0009 | |
| Soil texture | $x18$ | −0.0725 | | −0.0841 | |
| Relief degree of land surface | $x19$ | −0.0460 | | −0.0265 | |

Table 4 shows the relationship between the MCI and its impact factors. As for the nonagricultural process factors, in 2000 and 2015, the regression results showed that the influence of the four factors of the nonagricultural process on the MCI was almost negative. Especially for the nonagricultural industry, its influence coefficient changed from −0.0543 in 2000 to −0.6093 in 2015, demonstrating an obvious negative effect on the MCI. As for

cultivated land quality, in 2000, all the other impact factors were positively correlated with the MCI except natural disasters. However, in 2015, the effect of agricultural plastic film on the MCI became negative, and the density of pesticides showed a significant downward trend, from 9.7628 to 3.7945. As for the agriculturally intensive production factors, the impact coefficient of the grain yield per unit area, irrigation rate, and agricultural machinery per unit area were all positive, whereas the remaining factors were almost negative, but the negative effect of cultivated area per capita and grain yield per labor force on the MCI significantly declined by 2015. As for the qualities of the natural conditions, the results showed that the average annual precipitation and average annual temperature positively impacted the MCI; meanwhile, the soil texture and the relief degree of land surface negatively affected the MCI.

## 4. Discussion

### 4.1. Effects of Natural Conditions on MCI

Regarding natural conditions, hydrothermal conditions are the basic influencing factors for the multiple croppings of cultivated land, and the soil texture and the relief degree of the land surface also have important effects on the multiple croppings of cultivated land. Studies have shown that the realization of a multiple-cropping system depends largely on temperature and precipitation—sufficient accumulated temperature and rainfall are necessary to realize multiple croppings [48,67–69]. Affected by climate change, the northern limits of multiple-cropping systems have moved northward, and the projected area of cultivated land for multiple croppings may significantly expand during the 21st century in China [67]. Notably, the soil texture is closely related to soil aeration and water and fertilizer conservation. Poor soil texture inhibits the implementation of multiple croppings. Studies have confirmed that soil fertility is a key factor affecting the crop yield in the cropping system—notably, high soil fertility can increase the crop yield in a multiple-cropping system [59], but poor soil texture is the main reason a multiple-cropping system becomes ineffective [47]. Moreover, the land's relief degree is generally associated with multiple croppings: the higher the topography of the cultivated land, the more difficult it is to cultivate and achieve multiple-cropping systems [70]. Crops are suitable for cultivation in areas with relatively flat topography; high topographies can be hot spots for geological disasters, thus hindering the cultivation of crops. Additionally, areas with high topography generally have low levels of agricultural modernization, including the serious abandonment of cultivated land, which can dampen the MCI. However, technological developments can weaken the negative effect of topography on the MCI.

### 4.2. Adverse Effects of Nonagricultural Process on MCI

Nonagricultural processes in rural areas have an inhibitory effect on the increase of the MCI, especially the nonagricultural effects of the industry and population, which are the most direct factors that lead to the weakening of agricultural production subjects. Seeking higher economic benefits, the labor force engaged in agricultural production of the nonagricultural industry, with a large number of farmers working in cities and settling down as the main force to promote the process of urbanization [17,71,72]. Given the stable transformations in peasants' livelihoods, the rural labor force keeps reducing, and the problems of land abandonment and non-grain conversion become prominent [17,73], and the nonagricultural use of cultivated land has become one of the most typical trends of land use transitions in China [50,74], producing a negative impact on the MCI of cultivated land. In recent years, the Chinese government has focused on deploying major strategies to support the development of agriculture and rural areas, resulting in the adjustment of the agricultural structure, changing the morphology of cultivated land use, promoting the diversification of rural regional functions, and gradually diversifying the rural types. The rural development does not accomplish only an agricultural function; the secondary and tertiary industries rise gradually. The flow and agglomeration of various factors offer possibilities for the diversification of the rural industry. To some extent, this process

weakens the status of agriculture, makes the labor force originally left in the countryside turn to a new type of nonagricultural industry and then encourages the emergence of the nonagricultural industry and nonagricultural employment [70,75]. To prevent the lack of the main body of agricultural production, the study on the transformation of farmers' livelihoods and the multiple-cropping potential of cultivated land should be further strengthened, focusing on the identification of effective measures to encourage the return of rural labor force through micro-case studies to stabilize the main body of agricultural production.

*4.3. Effect of Intensive Agricultural Production on the MCI*

Intensive agricultural production promotes the MCI by improving the agricultural production efficiency—taking the grain yield as a measure of the agricultural economic benefits and agricultural modernization as a measure of the agricultural production efficiency, these items have a positive impact on multiple croppings. The grain yield per unit area directly affects the economic income of farmers growing grain, and the high agricultural income achieved by a high grain output can encourage farmers to engage in multiple croppings. However, improved multiple croppings of the cultivated land are limited by water resources; therefore, adequate irrigation is necessary to improve the MCI of the cultivated land significantly [29,54]. Here, it is helpful to note that the agricultural machinery represents the level of agricultural modernization. The higher the productivity and efficiency of the cultivated land with superior agricultural machinery conditions, the more inclined farmers are to carry out multiple croppings [55,76]. Our research found that farmers with more cultivated areas per capita may not be accordingly more enthusiastic about multiple croppings—this may be related to the small scales, insufficient production input, and weak labor force of some farming operations. In China, fertile lands in many mountainous areas have been abandoned, but the Chinese government is currently stepping up efforts to reform the land market, such as developing the rural land rental market to facilitate larger-scale farming operations and reduce the area of abandoned cultivated land [77,78].

*4.4. The Effect of Cultivated Land Quality on the MCI*

Cultivated land quality affects the morphology of cultivated land use, determines whether the land can be cultivated sustainable, and guarantees the long-term implementation of a multiple-cropping system. Natural disasters are a direct factor affecting the growth of multiple croppings. On the one hand, natural disasters objectively block the normal growth period of crops. On the other hand, natural disasters destroy the balance of soil and water, changing the spatial structure of the cultivated land, causing soil erosion and soil nutrient loss, resulting in a decline in the cultivated land quality, which will continue to affect the planting of the next crop. Multiple cropping requires high soil nutrient contents; therefore, large amounts of pesticides and fertilizers are required. The input of pesticides and fertilizers can improve the grain yield, ensure the production efficiency of multiple croppings, and make farmers willing to retain multiple-cropping behaviors. Notably, agricultural plastic film plays a very important role in the moisture and heat preservation of crops. The appropriate use of agricultural film can promote crop growth and is conducive to the multiple croppings of cultivated lands. However, the overuse and improper treatment of pesticides, fertilizers, and the film changes the soil's properties, causes agricultural nonpoint source pollution, and contributes to a number of environmental problems [79,80]. The continuation of this unscientific use will make it difficult to maintain multiple-cropping systems. What is important to take away here is that pesticides, fertilizers, and the film will continue to play vital roles in China's multiple-cropping system, which, as noted above, is of great significance to food security. Therefore, in the future, we need to balance the relationship between increasing the crop yields and reducing the environmental pollution while at once strengthening research on green agricultural systems, biodegradable film promotion, pesticide and fertilizer use efficiency, and alternative chemical fertilizers.

### 4.5. China's Cultivated Land Protection Policy and the MCI

This research showed that cultivated lands in China are facing the double pressure of area reduction and MCI reduction. Some studies support the idea that agricultural policies can stimulate cultivation, thus stabilizing the MCI [55,70]. China has been implementing the strictest cultivated land protection policies for many years, such as the policies of "taking grain as the highest priority", "red line of 1.8 billion mu of cultivated land in China", "basic cultivated land protection system", and "cultivated land requisition-compensation balance" [81]. Although the "red line of 1.8 billion mu of cultivated land" in China has not been broken, merely focusing on maintaining the total area of cultivated land has not prevented the loss and abandonment of a large portion of high-quality cultivated land. Many studies have shown that a large amount of high-quality cultivated land had been lost in the process of urbanization. Land development, land reclamation, land consolidation, and other measures lead to the occupation of high-quality cultivated land by low-quality cultivated land, and the existing policies for cultivated land have not managed to guarantee quality and quantity, resulting in the imbalance of cultivated land quality across the whole country [81–83]. How do the agricultural policies really guarantee food security? In addition to quantity, more attention should be paid to cultivated land quality, the farmer who cultivates the land, and the multiple-cropping potential of the cultivated land. It is important to tap the multiple-cropping potential of cultivated lands and make good use of the advantages of the regional MCI and incorporate the increased sowing area of the MCI into the assessment system of local officials to stimulate the enthusiasm of local governments to ensure food security.

### 4.6. Research Limitations

In this research, MODIS-NDVI was used as the data source to extract the MCI of the cultivated lands with the Twi-difference algorithm. Compared with the statistical data, remote sensing data can eliminate human disturbance and are more objective. However, the extraction of the MCI from remote sensing data may be affected by mixed pixels, which results in uncertainty. Due to the lack of field investigation in this paper, the accuracy of the extracted results cannot be verified. Further research will need to address this issue. As for the analysis of the factors and mechanisms influencing the MCI, given the limited availability of the data, this study selected the provincial scale for exploratory research. Although most of the selected indicators were based on rural statistical indicators, the provincial scale was too large, and the research results were uncertain. For example, the influence of important factors such as temperature and precipitation was positive, but the coefficient was close to 0, which can only reflect the general situation at the macro level. In the future, the scale of the research should be reduced to improve the accuracy of the results.

## 5. Conclusions

Using MODIS data, the spatial distribution and dynamic changes of the MCI in China in 2000, 2005, 2010, and 2015 were presented at different scales. Meanwhile, principal component regression was used to study the influencing factors. Our results suggest that the number of provinces with lower MCI increased gradually, and the dynamic changes in the MCI in the southern provinces were obvious at the provincial scale, especially in the MCI of the coastal provinces such as Fujian, Guangdong, Jiangxi, Zhejiang, and Shanghai, which declined significantly from 2000 to 2015. The mean MCI in Guangdong, Guangxi, Hainan, Henan, Anhui, and Jiangsu was high, whereas that of Heilongjiang, Inner Mongolia, Ningxia, and Qinghai was low. However, at the county scale, we found that the spatial distribution of the MCI differed from that at the provincial scale. The single-cropping index and MCI of China from 2000 to 2015 were roughly bounded by the "Taihang Mountains–Qinling Mountains–Hengduan Mountains"—the MCI was lower than 100% at the north of the boundary and higher than 100% at the south of the boundary. From 2000 to 2015, the dynamic changes of China's MCI were obvious, and the proportion

of the MCI with change values lower than 0 continued to rise. Notably, from 2005 to 2010 and 2010 to 2015, the MCI changed dramatically, mainly in the southeastern coastal areas; more specifically, the MCI rose in the former stage and dropped sharply in the latter stage.

We also examined and discussed the factors affecting the MCI. First, natural conditions are important factors in agriculture, among which hydrothermal conditions determine the upper limit of the MCI, and soil texture and the relief degree of the land surface inhibit the increase of the MCI. Second, nonagricultural processes adversely affect the MCI, especially the growth of the nonagricultural population and nonagricultural industry, which weaken the identity of farmers as the main body of agriculture and the rural agricultural production function. Third, the factors of intensive agricultural production, such as grain yield per unit area, irrigation rate, and agricultural machinery condition, promote the MCI by improving the agricultural production efficiency. Fourth, the cultivated land quality determines whether the land can be cultivated sustainably. For example, natural disasters cause water and soil imbalances. Meanwhile, pesticides, fertilizers, and a film can improve the grain yield and ensure the production efficiency of multiple croppings; however, their unscientific use will make farmlands unsustainable.

Given the continuous, rapid growth of nonagricultural processes, more research should be done on the transformations occurring in farmers' livelihoods, especially on the return of the rural labor force and the main body of agricultural production. Moving forward, it will be necessary for scholars and practitioners alike to balance the increases in crop production with the reductions in environmental pollution to grapple the increasingly severe agricultural ecological problems faced by China. Agricultural policies are of great significance to agricultural development; accordingly, moving forward, the policies should not only focus on the quantity of agricultural output but, also, on advancing the cultivated land quality, the farmer who cultivates the land, and the multiple-cropping potential of the cultivated land.

**Author Contributions:** Conceptualization, R.Y.; methodology, X.L. and J.W.; software, X.L. and J.W.; validation, R.Y. and X.L.; formal analysis, X.L.; investigation, X.L. and X.Z.; resources, Q.X. and R.Y.; data curation, Q.X.; writing—original draft preparation, X.L.; writing—review and editing, R.Y., X.L. and X.Z.; visualization, X.L. and X.Z.; supervision, Q.X. and R.Y.; project administration, Q.X.; funding acquisition, Q.X. and R.Y. All authors have read and agreed to the published version of the manuscript.

**Funding:** This research was financially supported by the National Natural Science Foundation of China (No. 41871177 and No. 41801088) and Natural Science Foundation of Guangdong Province(2018A0303130097).

**Institutional Review Board Statement:** Not applicable.

**Informed Consent Statement:** Not applicable.

**Data Availability Statement:** The data presented is primarily reflected in the article, more details are available on request from the corresponding author.

**Conflicts of Interest:** The authors declare no conflict of interest.

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
