# Peer review of "Measuring the Impact of the Multiple Cropping Index of Cultivated Land during Continuous and Rapid Rise of Urbanization in China: A Study from 2000 to 2015"

_land, doi:10.3390/land10050491_

Round 1
Reviewer 1 Report
they should point out the sources of tables, graphs and mapsAuthor Response
Dear reviewer:
Thank you for your comments on our manuscript entitled "Measuring the impact of the multiple cropping index of cultivated land during continuous and rapid rise of urbanization in China: A study from 2000—2015" (Manuscript ID: land-1190133). Your comments were highly insightful and enabled us to greatly improve the quality of our manuscript. Revisions in the manuscript are as shown in the resubmitted manuscript, and the responds to the reviewers’ comments are as follows (the replies are highlighted in blue).
No.1: They should point out the sources of tables, graphs and maps.
Response:
Firstly, we added data sources of tables, graphs and maps in the “2.3 Data sources” section, from line 267 to line 288 in the resubmitted manuscript. Details are as follows: First, MODIS-NDVI remote sensing image data and vector data in 2000, 2005, 2010, and 2015 on the spatiotemporal distributions of agricultural lands were provided by the Earth System Scientific Data Sharing Platform, Chinese Academy of Science (Data source of Figure 4). Second, the indexes of influencing factors included information on the economic, social, and natural conditions. The economic and social data, such as population, gross domestic product, cultivated land area, and grain output, were derived from the China Statistical Yearbook (2001, 2016), China Rural Statistical Yearbook (2001, 2016), China Population & Employment Statistics Yearbook (2001, 2015), and other statistical yearbooks for the provinces (Data source of Table 1, Figure 5). In particular, data for China’s agricultural population and non-agricultural population were only available until the end of 2014 (China Population & Employment Statistics Yearbook 2015); therefore, the data for 2015 were replaced by those from the end of 2014. Data on China’s cultivated land area were obtained from China’s Land and Resources Bulletin (Data source of Table 1, Figure 5). Third, the natural condition data, such as soil texture, annual average temperature, and annual average precipitation were obtained from the Resource and Environment Data Cloud Platform, Chinese Academy of Science (Data source of Table 1). Data sources for soil texture include the percentages of sand, silt, and clay. According to the China soil texture classification (1985) [64], the soil texture data were divided into three types: clay, loam, and sand. They were marked as 1, 2, and 3, respectively, in the regression analysis. Besides, the relief degree of land surface was derived from the research data of You et al. [65], who shared it in the Global Change Research Data Publishing & Repository (Data source of Table 1). Finally, the deflation index used to process economic data, the calculation process involved in the GDP, and the GDP index were taken from the China Statistical Yearbook (Data source of Table 1).
Secondly, we added a note to Figure 6–10 in the "2.2.1 Extraction methods of MCI" section, on line 215 in resubmitted manuscript. Details are as follows: Next, the provincial and county administrative units were used to carry on the statistics related to MCI, and the MCI means of Chinese counties and provinces were presented using the ArcGIS10.2 platform to prepare them for subsequent analysis (see Figure 6–10).
Once again, thank you very much for your constructive comments and suggestions which would help us both in English and in depth to improve the quality of the paper.
Kind regards,
Ren Yang
E-mail: yangren666@mail.sysu.edu.cn
Reviewer 2 Report
The manuscript titled “ Measuring the impact of the multiple cropping index of cultivated land during continuous and rapid rise of urbanization inChina: A study from 2000—2015. is a well-written manuscript. The paper could be very interesting for the reader of Land. I find the idea interesting and in line with the aim of the journal. I congratulate the author for a nice piece of work.. I also recommend to the authors improve their references by conducting a more extensive review of international literature. Particularly, in the introduction statements are not supported by the references selected by the authors. Below is my point-to-point analysis of the manuscript.
Statment „non-agricultural purposes [16] and the extraction of rural capital, laborers, and raw materials to promote industrialization [17]; accordingly, millions of rural agricultural laborers flowed into cities and non-agricultural industries” is not clear should be rewritten in a simple form.
Statment „Despite rural depopulation, rural settlements nearly tripled from 1967 to 2008, and almost all new non-agricultural land con- 68 sisted of farmland near villages [21], exacerbating the destruction of agricultural land.“ is not clear should be rewritten in a simple form.
Line No 86 should be researched instead carried out research on
Line no 99 should be a tremendous instead the greatest
Line no 122 should be researched instead carried out research on
Line no 139 should be increased instead increase
Line no 179 should be cultivated land quality instead the quality of cultivated land
Line no 504 should positive, whereas instead positive whereas,
Line no 547 should the pattern of cultivated land utilization cultivated land utilization patterns, instead of the pattern of cultivated land utilization,
Line no 568 should improve the MCI of cultivated significantly instead improve the MCI of cultivated significantly
Author Response
Dear reviewer:
Thank you for your comments on our manuscript entitled "Measuring the impact of the multiple cropping index of cultivated land during continuous and rapid rise of urbanization in China: A study from 2000—2015" (Manuscript ID: land-1190133). Your comments were highly insightful and enabled us to greatly improve the quality of our manuscript. Revisions in the manuscript are as shown in the resubmitted manuscript, and the responds to the reviewers’ comments are as follows (the replies are highlighted in blue).
No.1: I also recommend to the authors improve their references by conducting a more extensive review of international literature. Particularly, in the introduction statements are not supported by the references selected by the authors.
Response:
According to the suggestion, we checked the statement in the “1 Introduction” section, rewrote some of the content, and added some appropriate references. For example:
“On the one hand, MCI is an important measure for evaluating China’s food security, as its increase is essential to meet the growing demand for food [47,48]. On the other hand, MCI is an important evaluation index of land use transition research [49]. MCI researches the changes of recessive morphologies of land use transition, and reflects the land management model by measuring the intensity level of cultivated land use [50,51]. Therefore, it is of great practical significance to research MCI and its influencing factors to support national policy formulation in China [28,52,53].” (Line no 107–113).
“Cultivated land is an important land type in the research of rural land use transition, and it is closely related to human production and life. Cultivated land use reflects the evolution of human-environment relationship in rural areas, and also reflects the current situation and problems of China's agriculture and rural society.” (Line no 135–138).
No.2: Statment „non-agricultural purposes [16] and the extraction of rural capital, laborers, and raw materials to promote industrialization [17]; accordingly, millions of rural agricultural laborers flowed into cities and non-agricultural industries” is not clear should be rewritten in a simple form.
Response:
We rewrote this statement in the “1 Introduction” section, from line 51 to 54 in the resubmitted manuscript. Details as follows: Between the 1950s and 1990s, China’s urbanization caused the rapid transition of cultivated land use, many cultivated land transformed into non-agricultural use, and rural areas served urban development [16,17]; accordingly, millions of rural agricultural laborers flowed into cities and non-agricultural industries.
No.3: Statment „Despite rural depopulation, rural settlements nearly tripled from 1967 to 2008, and almost all new non-agricultural land con- 68 sisted of farmland near villages [21], exacerbating the destruction of agricultural land.“ is not clear should be rewritten in a simple form.
Response:
We rewrote this statement in the “1 Introduction” section, from line 60 to 63 in the resubmitted manuscript. Details as follows: Despite rural depopulation, the land area of rural settlements did not decrease correspondingly. On the contrary, rural settlements area nearly tripled from 1967 to 2008, and most of these increased areas occupied agricultural land, exacerbating the destruction of agricultural land [21].
No.4. Line No 86 should be researched instead carried out research on
Line no 99 should be a tremendous instead the greatest
Line no 122 should be researched instead carried out research on
Line no 139 should be increased instead increase
Line no 179 should be cultivated land quality instead the quality of cultivated land
Line no 504 should positive, whereas instead positive whereas,
Line no 547 should be cultivated land utilization patterns, instead of the pattern of cultivated land utilization,
Line no 568 should improve the MCI of cultivated significantly instead improve the MCI of cultivated significantly
Response:
We checked each suggestion point-to-point and made corresponding modifications in the resubmitted manuscript. Details as follows: “Scholars from different countries have researched MCI in different regions of the world…” (Line no 80). “these studies found that Latin America had a tremendous potential…” (Line no 91). “…domestic scholars actively researched MCI,” (Line no 114). “with an increased rate of about 20.2%” (Line no 128). “Third, cultivated land quality affects the increase of MCI” (Line no 161). “and agricultural machinery per unit area were all positive, whereas the remaining factors were almost negative” (Line no 464). “therefore, adequate irrigation is necessary to improve the MCI of cultivated land significantly” (Line no 523).
Once again, thank you very much for your constructive comments and suggestions which would help us both in English and in depth to improve the quality of the paper.
Kind regards,
Ren Yang
E-mail: yangren666@mail.sysu.edu.cn
Reviewer 3 Report
Dear all,
Many thanks for the invitation to review this potential article.
Generally, this has relevance and presents scientific soundness.
Yet, some minor improvements (mainly regarding the format) should be conducted; as example:
-Figure 2 caption - is on small letters
-confirm if the document is on the MDPI proper formatting
- several figures are too small and difficult to read - instead of having so many, some should be sent to Appendix in a larger size
Best,
Author Response
Dear reviewer:
Thank you for your comments on our manuscript entitled "Measuring the impact of the multiple cropping index of cultivated land during continuous and rapid rise of urbanization in China: A study from 2000—2015" (Manuscript ID: land-1190133). Your comments were highly insightful and enabled us to greatly improve the quality of our manuscript. Revisions in the manuscript are as shown in the resubmitted manuscript, and the responds to the reviewers’ comments are as follows (the replies are highlighted in blue).
No.1: Figure 2 caption - is on small letters
Response:
We have revised it in the “2.1 Research structure” section. Details as follows: Figure 2. The research framework concerning the influencing factors. (Line no 155).
No.2: Confirm if the document is on the MDPI proper formatting
Response:
We have checked the format of the document and modified some error formats. For example, all “Fig” of figure caption revised to “Figure”, and also modified the incorrectly formatted references.
No.3: Several figures are too small and difficult to read - instead of having so many, some should be sent to Appendix in a larger size
Response:
We are very grateful to this suggestion, and we have modified the size of each figure for better reading. For example:
Figure 5. China’s cultivated land area and grain yield, 2000-2015.
Figure 6. The value of MCI in Chinese provinces, 2000–2015.
Figure 7. The changes in the value of MCI in Chinese provinces from 2000–2005, 2005–2010, and 2010–2015.
Once again, thank you very much for your constructive comments and suggestions which would help us both in English and in depth to improve the quality of the paper.
Kind regards,
Ren Yang
E-mail: yangren666@mail.sysu.edu.cn